# An Adaptive Deep RL Method for Non-Stationary Environments with Piecewise Stable Context

**Xiaoyu Chen**$^{*\diamond}$  **Xiangming Zhu**$^{*\clubsuit}$  **Yufeng Zheng**$^{\spadesuit}$  **Pushi Zhang**$^{\ddagger}$

**Li Zhao**$^{\dagger\ddagger}$  **Wenxue Cheng**$^{\ddagger}$  **Peng Cheng**$^{\ddagger}$  **Yongqiang Xiong**$^{\ddagger}$

**Tao Qin**$^{\ddagger}$  **Jianyu Chen**$^{\diamond}$  **Tie-Yan Liu**$^{\ddagger}$

$\diamond$ **Tsinghua University**
$\clubsuit$ Shanghai Jiao Tong University
$\spadesuit$ University of California, Berkeley
$\ddagger$ Microsoft Research Asia

## Abstract

One of the key challenges in deploying RL to real-world applications is to adapt to variations of unknown environment contexts, such as changing terrains in robotic tasks and fluctuated bandwidth in congestion control. Existing works on adaptation to unknown environment contexts either assume the contexts are the same for the whole episode or assume the context variables are Markovian. However, in many real-world applications, the environment context usually stays stable for a stochastic period and then changes in an abrupt and unpredictable manner within an episode, resulting in a segment structure, which existing works fail to address. To leverage the segment structure of piecewise stable context in real-world applications, in this paper, we propose a ***Se**gmented **C**ontext **B**elief **A**ugmented **D**eep (SeCBAD)* RL method. Our method can jointly infer the belief distribution over latent context with the posterior over segment length and perform more accurate belief context inference with observed data within the current context segment. The inferred belief context can be leveraged to augment the state, leading to a policy that can adapt to abrupt variations in context. We demonstrate empirically that SeCBAD can infer context segment length accurately and outperform existing methods on a toy grid world environment and MuJoCo tasks with piecewise-stable context.

## 1 Introduction

Deep reinforcement learning has achieved great success in a wide range of challenging environments such as Atari games (Mnih et al., 2013; Bellemare et al., 2012; Hessel et al., 2017) or continuous control tasks (Schulman et al., 2015, 2017a). However, in stark contrast with this trend, applying RL to real-world applications remains a great challenge. In most real-world settings, there could be variations in environmental factors, such as changing terrains in robotic tasks, fluctuated bandwidth in congestion control, and dynamic traffic patterns in autonomous driving. We refer to such environment factors as *situation* or *context*. The changes in context are not neglectable since context usually has a substantial impact on transition and reward functions. When the context is fixed and known to us, the problem is stationary and easier to solve. However, in most realistic settings, the context is usually dynamic within an episode and unknown to us at test time. Therefore, detecting and adapting

---

$^{*}$Equal contribution. This work is conducted at Microsoft.
$^{\dagger}$Corresponding author.

36th Conference on Neural Information Processing Systems (NeurIPS 2022).

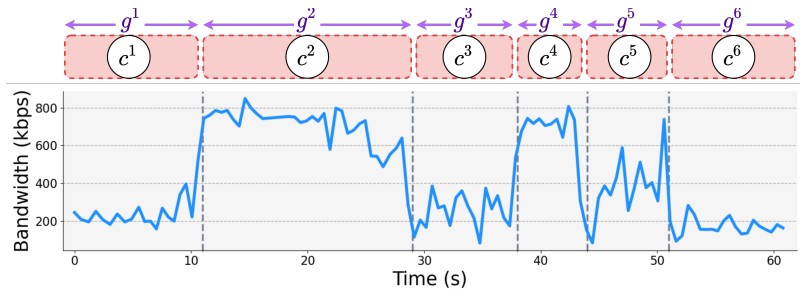

Figure 1: A typical trace of network available bandwidth. The trace can be approximately divided into several segments of different length $g^i$, for $i \in \{1, 2, \cdots, 6\}$. The network condition changes abruptly at the end of each segment, while in each segment the network condition stays stable.

to variations in context is very important for RL agents to make a real impact in a wide range of real-world applications.

In real-world environments with varied unknown contexts, we find a typical evolving context pattern particularly interesting. The context $c$ usually stays the same for a stochastic period until it changes abruptly and unpredictably into another context value $c'$ which is sampled i.i.d. from some prior context distribution. What's more, we usually do not have access to context $c$ directly but instead have access to some noisy observation $x_t$ sampled from some distribution $p(x_t|c)$ only at training time (e.g., as auxiliary information from the simulator). We take fluctuated bandwidth in congestion control as an example. In Figure 1, we show a typical trace of network available bandwidth. The fluctuated bandwidth is usually modeled by multiple non-overlapping segments (Akhtar et al., 2018; Zhang and Duffield, 2001). Within each segment, the network condition is stationary, and thus the bandwidth approximately follows the same distribution. At the end of a segment, the network condition changes, and the bandwidth changes abruptly into another distribution. Here the context $c$ represents the parameters that determine the distribution of fluctuated bandwidth (network condition), while the observations $x_t$ represents the observed bandwidth which follows the above distribution $p(x_t|c)$. While the contexts $c$ are piecewise-stable, there could be slight variations in $x_t$ within each segment. This piecewise-stable pattern of context dynamics is of particular interest to us since it can capture a wide range of stochastic context processes in real-world applications, such as changing terrains in robotic tasks and fluctuating bandwidth in congestion control. Since the change in context is abrupt and unpredictable, we cannot predict the future context in advance. The best we can do is detect and adapt to changes when they happen.

Although adaptation to varied unknown contexts has been studied under non-stationary RL and meta RL, few existing works look into piecewise stable context with abrupt changes within an episode. Most works in meta RL as task inference (Rakelly et al., 2019; Zintgraf et al., 2020; Zhao et al., 2020; Poiani et al., 2021) and some works in non-stationary RL (Chandak et al., 2020a,b; Xie et al., 2021) assume the context stays the same for the whole episode and infer the context based on the entire episode (c.f. Figure 2(a)), therefore cannot quickly adapt to context changes within an episode. Other works on non-stationary RL assume intra-episode context changes and model $c_t$ at each time step, but few study the piecewise-stable context as we do. Nagabandi et al. (2018) directly predict the context and thus cannot capture the prior that the context tends to stay the same for a stochastic period. Feng et al. (2022); Ren et al. (2022) model Markovian discrete context for each time step (c.f. Figure 2(b)), therefore failing to model the non-markov property of context and the prior over context segment length in our setting. Compared with existing works, our setting is distinctive since the segment structure is latent to us (c.f. Figure 2(c)). Therefore, the unique challenge in our setting is that we need to infer the segment structure, which can be further leveraged to infer belief context by only incorporating the relevant observed data in the current segment.

This paper studies how to infer segment structure to detect abrupt context changes and infer belief context accordingly to adapt to the piecewise-stable context in RL environments. We first introduce latent situational MDP which models RL environments with the stochastic *situation/context* process (Section 2). Then, we introduce how to infer the belief context from observed data. To address the challenge above, we propose to infer context segment structure and belief context jointly from observed data (Section 3.1). Then we augment the state with inferred belief context so that the RL agent can automatically trade-off between acting optimally conditioned on inferred context and gathering

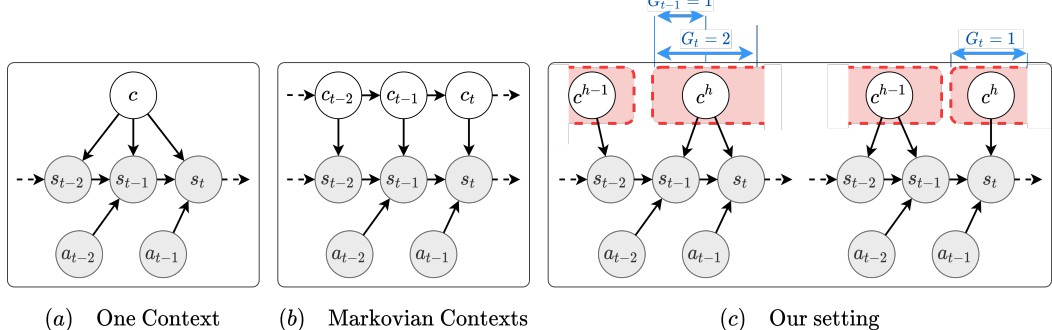

(a)   One Context          (b)   Markovian Contexts          (c)   Our setting

Figure 2: Probabilistic Graphical Models (PGMs) for different problem settings, with shaded circles for observable variables and white circles for latent variables. (a) One context: assume the context remains unchanged for the whole episode. (b) Markovian context: assuming the context is Markovian for each time step. (c) Our setting: the context remains unchanged in each segment, but the segment structure, illustrated by the red segment is unknown and needs to be inferred. We show two possible examples of PGM corresponding to $G_t = 2$ ((c), left) and $G_t = 1$ ((c), right), where $G_t$ measures the length of the current segment up to time step $t$.

more information about the current context (Section 3.2). Finally, we combine the training objectives for RL and inference and present all the details for our proposed deep RL algorithm (Section 3.3). We evaluate our algorithm on a gridworld environment with dynamic goals and MuJoCo tasks (Todorov et al., 2012) with varied contexts. Experiments demonstrate that our algorithm can quickly detect and adapt to abrupt changes in piece-wise stable contexts and outperform existing methods(Section 4). Our contributions can be listed as follows:

- We introduce latent situational MDP with piecewise-stable context, which can capture a wide range of real-world applications (Section 2).
- We propose SeCBAD, an adaptive deep RL method for non-stationary environments with piecewise-stable context. Our method can infer the context segment structure and the belief context accordingly from observed data, which can be leveraged to detect and adapt to context changes (Section 3).
- Experiments on a gridworld environment and Mujuco tasks with piecewise-stable context demonstrate that our method can quickly detect and adapt to abrupt context changes and outperform existing methods (Section 4).

## 2   Problem Formulation

In this section, we define a *latent situational Markov Decision Process (LS-MDP)* as a tuple $M = (\mathcal{S}, \mathcal{A}, \mathcal{C}, \mathcal{X}, G, T, R, \gamma)$, where $\mathcal{S}$ is the set of states, $\mathcal{A}$ is the set of actions, and $\gamma \in (0, 1]$ is the discount factor. To formulate the variations in environment factors, we introduce $\mathcal{C}$, the set of latent contexts, and $\mathcal{X}$, the set of observable contexts. $\mathcal{C}$ refers to the set of underlying hidden contexts that contains all necessary information but are left unobservable to the agent, while $\mathcal{X}$ refers to the set of contexts with only partial information. $G$ refers to the segment length which will be described in detail in the next paragraph. $\mathcal{X}$ are observable only during training and unobservable during deployment. In an example environment setting where a robot walking over changing terrain, $\mathcal{C}$ represents the terrain features containing perfect information, and $\mathcal{X}$ represents noisy and imperfect information like mechanical metrics of the current step from virtual sensors in a simulator and thus is only accessible during training.

In our setting, we focus on the case where the environmental changes are abrupt and irregular, in contrast to the smoothly changing assumption on context/task in existing works in non-stationary RL (Chandak et al., 2020a,b). To better model the generative process of the contexts, we introduce the segment length $G$. Each episode is composed of several stationary segments with different segment lengths. For the ease of notation, we also introduce $G_t$, which measures the length of the current segment up to time step $t$. Then, the generative process can be described as follows: at the beginning of the $h$-th segment, the environment samples $G^h \sim p_G(G)$ and $c^h \sim p_c(c)$ from prior distributions $p_G$ and $p_c$. Then, in the next $G^h$ steps, the latent context $c^h$ remains unchanged. For each time step $t$

in this segment, the current segment length accumulates as $G_t = G_{t-1} + 1$ (we define $G_t = 1$ for the first time step in this segment), and the current latent context satisfies that $c_t = c^h$. The agent can observe $x_t \sim p(x_t|c_t)$ for each time step if during training. The stationarity lasts until the end of the current segment, then the environment resamples $G^{h+1}$ and $c^{h+1}$, and the process repeats. We show two examples of graphical models of our setting given different $G_t$ in Figure 2(c).

The transition function $T : p(s_{t+1}|s_t, a_t, c^h)$ and the reward function $R : p(r_t|s_t, a_t, c^h)$ are all conditioned on current latent context $c^h$. Therefore the changes in $\mathcal{C}$ lead to the changes in transition and reward functions. At test time, $\mathcal{X}$ is no longer accessible, so we need to infer context changes from observed transitions and rewards. Since the latent context $\mathcal{C}$ remains unobservable and changes silently, the environment is no longer stationary for the agents. To act optimally, it is important to keep track of the environment to recognize changes in time, and rapidly adapt to those changes.

## 3 Methodology

In this section, we present our *Segmented Context Belief Augmented Deep (SeCBAD)* RL method and elaborate on how SeCBAD solves the challenges discussed above. Our method consists of two main components:

- Joint inference of the belief distribution over the latent context and the segment structure from observed data.
- Policy optimization with inferred belief context under the belief MDP framework.

We first introduce the latent context inference part in Section 3.1, especially how to infer the segment structure jointly with belief context and leverage the segment structure to remove irrelevant observed data. After the belief context is approximated, it is then incorporated into the state as the input of the policy. We detail the policy optimization part under the belief MDP framework in Section 3.2. And finally, in Section 3.3, we describe how these parts constitute a practical algorithm.

### 3.1 Joint Inference for Belief Context and Segment Structure

In this part, we perform joint inference over latent context $c_t$ and current segment length $G_t$ from observed trajectory $\tau_{1:t}$, so as to remove irrelevant data in $\tau_{1:t}$ for belief context inference. This can be formally expressed by the following equation:

$$p(c_t, G_t|\tau_{1:t}) = p(c^{t-G_t+1:t}|G_t, \tau_{t-G_t:t})p(G_t|\tau_{1:t}) \tag{1}$$

where $\tau_{t_0:t} = (s_{t_0}, a_{t_0}, r_{t_0} \cdots, s_{t-1}, a_{t-1}, r_{t-1}, s_t)^3$, and $c^{t-G_t+1:t}$ refers to the latent context for the whole segment from $t - G_t + 1$ to $t$. In Equation 1, we separately estimate the posterior $p(c^{t-G_t+1:t}|G_t, \tau_{t-G_t:t})$ of latent context given known segment structure $G_t$, and the posterior of segment length $p(G_t|\tau_{1:t})$.

#### 3.1.1 Approximate Inference for Latent Context under Known Segment Structure

In this part, we focus on estimating $p(c^{t-G_t+1:t}|G_t, \tau_{t-G_t:t})$, which is the posterior of the latent context under known segment structure $G_t$. We use the variational inference framework to approximate the true posterior. To be specific, we use an posterior inference network $q_\phi$ to infer the belief context in segment $[t - G_t + 1 : t]$: $q_\phi(c^{t-G_t+1:t}|G_t, \tau_{t-G_t:t})$. The variational lower bound for the log-likelihood of the current segment is given by:

$$\log p(\tau^X_{t-G_t:t}|a_{t-G_t:t-1}, s_{t-G_t}, G_t)$$
$$\geq \mathbb{E}_{q_\phi(c^{t-G_t+1:t}|G_t, \tau_{t-G_t:t})} \left[\log p_\theta(\tau^X_{t-G_t:t}|G_t, c^{t-G_t+1:t}, a_{t-G_t:t-1}, s_{t-G_t})\right]$$
$$- \mathbf{D}_{KL}(q_\phi(c^{t-G_t+1:t}|G_t, \tau_{t-G_t:t})\|p(c^{t-G_t+1:t})) := \mathcal{J}^t_{Model}(G_t) \tag{2}$$

where $p_\theta$ denotes the decoder and $\tau^X_{t-G_t:t} = (\tau_{t-G_t:t}, x_{t-G_t+1:t})$. For the detailed derivation, please see Appendix A.1.

The reconstruction term of Equation 2 can be factorized as:

$$\log p_\theta(\tau^X_{t-G_t:t}|G_t, c^{t-G_t+1:t}, a_{t-G_t:t-1}, s_{t-G_t})$$
$$= \sum_{i=t-G_t+1}^{t} \log p_\theta(x_i, s_i, r_{i-1}|G_t, c^{t-G_t+1:t}, s_{i-1}, a_{i-1}) \tag{3}$$

---

[3]This definition makes $\tau_{t-G_t:t}$ only contain observed data that belong to the current segment.

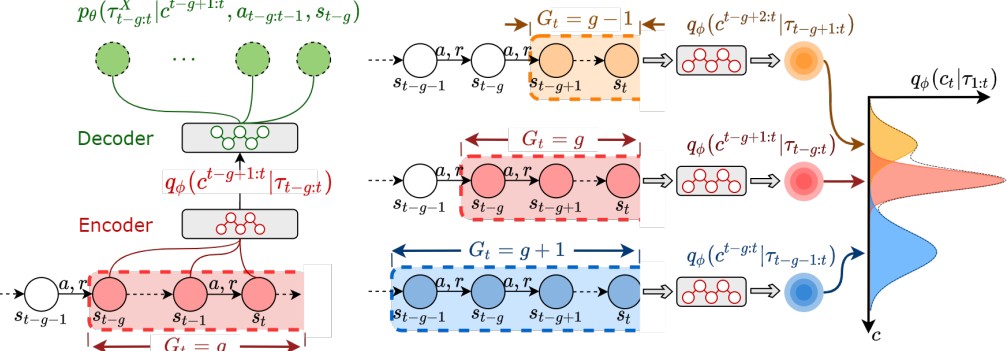

(a)  Encoder-Decoder Architecture  (b)  Belief Context as a Mixture Distribution

Figure 3: An overview of SeCBAD. (a) Encoder-decoder architecture for belief context inference given current segment length $G_t = g$: the encoder $q_\phi$ takes the recent $g$ steps in the current red segment as input, and infer the belief context $q_\phi(c^{t-g+1:t}|\tau_{t-g:t})$, and the decoder takes in sampled context $c$ from $q_\phi$ and decode the trajectory segment $\tau^X_{t-g:t}$ (the green nodes). (b) Belief context $q_\phi(c_t|\tau_{1:t})$ as mixture distribution by considering belief context $q_\phi(c^{t-g+1:t}|\tau_{t-g:t})$ for all possible structures according to the posterior $p(G_t|\tau_{1:t})$.

which is the sum of log-probablity of transitions under context sampled from $q_\phi$. The term $\mathbf{D}_{KL}(q_\phi||p)$ is the KL-divergence between our variational posterior $q_\phi$ and the prior over the belief context. For the prior $p(c^{t-G_t+1:t})$, we use previous posterior at timestep $t-1$ if the context remains unchanged at timestep $t$, or $\mathcal{N}(0, I)$ otherwise.

Unlike most previous methods which assume invariant context within an episode or markovian context at each time step, our method 1) takes only observed data within current segment $\tau_{t-G_t:t}$ as input for encoder $q_\phi$, 2) reconstructs only data within current segment $\tau^X_{t-G_t:t}$ at the output for decoder $\log p_\theta$. Our method is naturally motivated by the piecewise-stable assumption on context dynamics. By removing irrelevant data outside the current segment that are generated by past unassociated context, our method can estimate the current latent context more accurately.

### 3.1.2  Iterative Inference for the Segment Length

In this part, we focus on estimating $p(G_t|\tau_{1:t})$, which is the posterior for current segment length, given $p(c^h|\tau_{t-G_t:t})$, the posterior of context under given segment structure. To compute this posterior distribution, we first compute the joint distribution $p(G_t, \tau_{1:t})$ recursively based on $p(G_{t-1}, \tau_{1:t-1})$ as follows[4]:

$$p(G_t = i, \tau_{1:t}) = \sum_{k=1}^{t-1} p(G_{t-1} = k, \tau_{1:t-1}) \cdot p(G_t = i|G_{t-1} = k) \cdot p(s_t, a_{t-1}, r_{t-1}|\tau_{t-i:t-1}) \quad (4)$$

The three major components in Equation 4 in turn are the previous joint distribution, the evolution prior, and the observation probability. The previous joint distribution is iteratively provided at the beginning of each time step. The evolution prior $p(G_t = i|G_{t-1} = k)$ measures the prior knowledge on the segment length $G_t$ given the segment length of the previous time step $G_{t-1} = k$, where either $G_t = G_{t-1} + 1$ or $G_t = 1$ holds.

As for the observation probability which is the third term of Equation 4, it measures how likely the observations show up given the history of the segment. To be specific, we have [5]

$$p(s_t, a_{t-1}, r_{t-1}|\tau_{t-i:t-1}) = K\mathbb{E}_{p(c^{t-i+1:t}|G_t=i,\tau_{t-i:t-1})}\Big[p(s_t, r_{t-1}|s_{t-1}, a_{t-1}, c^{t-i+1:t})\Big] \quad (5)$$

---

[4]For notation simplicity, we omit the condition $G_t = i$ in the term $p(s_t, a_{t-1}, r_{t-1}|\tau_{t-i:t-1})$, i.e. $t-i$ is the start of the segment.

[5]In Equation 5, $K = p(a_{t-1}|s_{t-1})$ is a constant with respect to $i$ and has no impact on the posterior $p(G_t = i|\tau_{1:t})$.

In Equation 5, the term $p(s_t, r_{t-1}|s_{t-1}, a_{t-1}, c^{t-i+1:t})$ estimates the data likelihood for the next state-reward pair, where $c$ is drawn from the posterior distribution given the data in the segment before timestep $t$. To compute the RHS of Equation 5, we sample from $q_\phi(c^{t-i+1:t-1}|G_t = i, \tau_{t-i:t-1})$[6] which is the belief context inferred from the whole history of the segment before timestep $t$, and use the sampled $c$ the decoder to compute the data likelihood.

Given the joint distribution, the posterior distribution of $G_t$ can be derived as

$$p(G_t = i|\tau_{1:t}) = \frac{p(G_t = i, \tau_{1:t})}{\sum_l p(G_t = l, \tau_{1:t})} \tag{6}$$

We can also incorporate observable contexts $x$ in the observation probability to improve accuracy during training. See Appendix A.3 for more details.

### 3.1.3 Belief Context as a Mixture Distribution

Given inferred posterior of $G_t$ in Section 3.1.2, the belief of the latent context at time step $t$ can be derived as

$$b_t(c) = q_\phi(c_t|\tau_{1:t}) = \sum_{g_t} q_\phi(c^{t-g_t+1:t}|G_t = g_t, \tau_{t-g_t:t})p(G_t = g_t|\tau_{1:t}) \tag{7}$$

This mixed probability of $c^{t-g_t+1:t}$ which has taken all possible segment structures into consideration, can now represent the current belief $b_t$ of the latent context $c$.

### 3.2 Policy Optimization with Belief Context

Inspired by the belief MDP (Kaelbling et al., 1998), Bayes Adaptive MDP (BAMDP) (Duff, 2002) and recent works on meta RL as task inference (Zintgraf et al., 2020), we incorporate the inferred belief context into the augmented state. At each time step $t$, the belief latent context is approximated via $q_\phi$ using Equation 7. Therefore, we define the augmented state as $(s, b) \in \mathcal{S} \times \mathcal{B}$, where $\mathcal{S}$ is the same state space as in LS-MDP and $\mathcal{B}$ is the set of belief latent contexts. Accordingly, we have transition $T^b(s_{t+1}, r_t, b_{t+1}|s_t, a_t, b_t) = p(s_{t+1}, r_t|s_t, a_t, b_t)p(b_{t+1}|s_t, a_t, r_t, s_{t+1}, b_t)$ and reward $R^b(r_t|s_t, b_t, a_t)$ This definition brings advantages in the sense that the information gathering and exploitation tradeoff is no longer a problem under such augmented states, since the transition and reward functions are no longer conditioned on exact $c$. Now, the policy is defined as $\pi(a|s, b)$ a mapping from the augmented state space to the action space, and the agent's objective is to maximize

$$J_{RL} = \mathbb{E}_{s_0, b_0, \pi, T^b} \left[ \sum_{t=0}^{H} \gamma^t R^b(r_t|s_t, b_t, a_t) \right]. \tag{8}$$

### 3.3 Algorithm and Implementations of SeCBAD

In this section, we describe the overall algorithm and implementation details of SeCBAD. See Figure 3 for an overview of our framework. As shown in Figure 3(a), we use a GRU (Cho et al., 2014) parameterized by $\phi$ as the recurrent encoder $q_\phi$, and distributions in latent context space is assumed to be diagonal Gaussians with mean and variance parameterized by $q_\phi$. The decoder includes transition model $p_\theta(s_{t+1}|s_t, a_t, c)$, reward model $p_\theta(r_t|s_t, a_t, c)$ and observable context model $p_\theta(x_t|c)$. The output of all the decoders are Gaussian distributions with mean parameterized by feed-forward neural networks and fixed identity covariance. Then, we estimate $p(G_t|\tau_{1:t})$ using $q_\phi$ and $p_\theta$ as described in Section 3.1.2. As shown in Figure 3(b), we combine the belief context based on different segment according to $p(G_t|\tau_{1:t})$ to get the belief $b_t(c) = q_\phi(c_t|\tau_{1:t})$, where one approach is to provide a total of $t$ mean, covariance and weights as policy input. For simplicity, we choose $G_t^*$ with highest probability in $p(G_t|\tau_{1:t})$ and use the corresponding $q_\phi(c^{t-G_t^*+1:t}|\tau_{t-G_t^*:t})$ as the belief. Empirically, we find this approximation leads to little performance loss. We build our RL algorithm on the top of PPO (Schulman et al., 2017b) to learn the policy $\pi_\psi(a_t|s_t, b_t(c))$, where $\psi$ denotes the parameters in the actor and the critic. We use the objective described in Section 3.2 to optimize the policy.

---

[6]When $i \geq 2$, we can use the belief $q_\phi(c^{t-i+1:t-1}|G_{t-1} = i - 1, \tau_{t-i:t-1})$ to approximate the posterior $p(c^{t-i+1:t}|G_t = i, \tau_{t-i:t-1})$. For the case where $i = 1$, the posterior distribution $p(c^{t-i+1:t}|G_t = i, \tau_{t-i:t-1})$ is actually the prior distribution $p_c(c) = \mathcal{N}(0, I)$, and we can sample $c$ from the prior distribution.

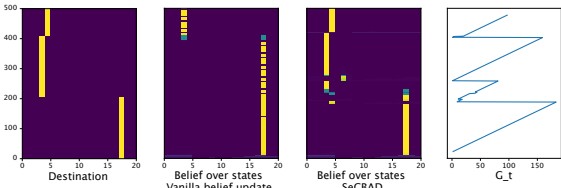

Figure 4: Experiment on grid world with dynamic goal state. We show a case study on the inferred belief. The leftmost figure depicts the ground truth goal position, and the next two figure depicts the belief estimated by vanilla inference and SeCBAD. where the color of one grid measures how the agent believes $s^*$ locates at that grid. The rightmost figure is $G_t^*$ estimated by SeCBAD. The result shows that our belief matches the ground truth more closely.

During deployment, $q_\phi$ and $p_\theta$ are fixed. We first compute the segment posterior $p(G_t|\tau_{1:t})$ using encoder $q_\phi$ and the transition and reward model $p_\theta$. Then, we estimate the belief $b_t(c)$ and feed it into the policy as input. For more implementation details, please refer to the Appendix A.6.

## 4 Experiments

In this section, we empirically evaluate SeCBAD on two tasks. We first demonstrate our algorithm in a grid world environment in Section 4.1 to illustrate the significance of incorporating segment structure during inference. For large-scale experiments, we test the proposed algorithm in Section 4.2. The results of multiple challenging tasks show that SeCBAD outperforms baselines in terms of performance and sample efficiency. In Section 4.3, we further provide case studies of agent behaviors and learned latent to gain more insights into the results.

### 4.1 Motivating Example

In this experiment, we use a motivating example to show that it is vital for the agent to consider the segment structure to adapt to variations of unknown environment contexts. The agent is in a $4 \times 5$ grid world containing 1 goal state $s^*$ and 19 other states. In $s^*$, the agent can gain $+1$ reward with probability $p_0$, and 0 reward with probability $1 - p_0$. For other states, the agent can gain $+1$ reward with probability $p_1$ and 0 reward with probability $1 - p_1$, where $p_0 > p_1$. The actions include up, down, left, right, and none. The specific location of $s^*$ is unobservable to the agent, and may change after a stochastic period.

We use the analytical belief update rule instead of variational inference in this experiment. To be specific, we assume the $q_\phi$ and $p_\theta$ is known and fixed and then infer $p(G_t|\tau_{1:t})$ and the belief $b_t(c)$, which represents the belief distribution of the location of $s^*$. The belief update rule is detailed in Appendix A.5. With the analytical belief update rule, a theoretically optimal belief $b_t$ is computed for each environment step $t$. We concatenate $(s_t, b_t)$ together as the augmented state. We may expect that a policy trained upon this augmented state will achieve better performance if $b_t$ is an accurate enough guess to the true location of the goal state. In such a way, we aim to focus on the effectiveness of inferring segment structure, in terms of how inferring segment structure can arrive at a meaningful belief state. We compare our algorithm with vanilla inference, where the agent updates belief without considering segment structure, like in Zintgraf et al. (2020); Rakelly et al. (2019). For both of two routines of updating belief state, we let the agent move along a same path. By moving along a same path, we mean the agent starts from the same location and then take the same action in each environment step for two belief update rules.

Figure 4 illustrates how SeCBAD and the vanilla inference baseline update their belief over states. It is shown that the belief $b_t(c)$ of SeCBAD closely matches with the actual goal state $s^*$. Also, compared with SeCBAD, the vanilla inference baseline needs more steps to detect the changes since it needs to correct the deviated prior belief. This result supports that incorporating segment structure into inference like SeCBAD is significant for estimating accurate belief in fast varying environments.

### 4.2 Locomotion Control Tasks with Varying Contexts

In this section, we show that SecBAD is able to tackle more complex tasks by testing the algorithm on several challenging control tasks with varying contexts. We conduct the experiments on modified MuJoCo (Todorov et al., 2012) tasks. These environments are commonly used in previous works

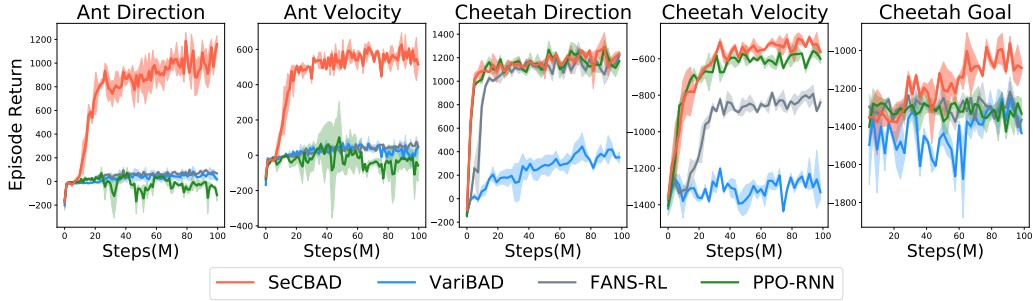

Figure 5: Performance curves on 5 MuJoCo environments with variations in contexts. SeCBAD achieves better performance and sample efficiency in various challenging control tasks.

(Zintgraf et al., 2020; Rakelly et al., 2019), and we further modify the contexts to provide challenges corresponding to LS-MDP as mentioned in Section 2. As for Ant Direction and Cheetah Direction, the policy needs to move towards the target direction as fast as possible. As for Ant Velocity and Cheetah Velocity, the policy then needs to additionally move at the speed of the target velocity besides the target direction. For Cheetah Goal, the policy needs to navigate to varying goals and stay there. For Ant environments, the direction and velocity are specified on a 2-dimensional plane, and for half cheetah environments, the direction and velocity are 1-dimensional. In these environments, the contexts refer to the target velocity, target direction, or location of the goal. To make it more challenging, we further add noises to contexts within each segment. Please refer to Appendix A.6 for more details. As for baselines, we select one representative algorithm for each type in Figure 2 to compare since to the best of our knowledge, few existing works study the same setting as we do. We use VariBAD (Zintgraf et al., 2020) to represent those methods assuming one context in the episode, and use FANS-RL (Feng et al., 2022) to represent those methods assuming Markovian contexts. Since LS-MDP is a special case of POMDP, we include a POMDP baseline PPO-RNN (Hausknecht and Stone, 2015) for comparison. The results are provided in Figure 5.

The experiment results empirically show that SeCBAD achieves superior performance and sample efficiency on challenging control tasks. As illustrated in Figure 5, SeCBAD achieves higher scores than baselines. We provide some insights into the results. For the method assuming that contexts stay the same within an episode, (Zintgraf et al., 2020) uses the learned latent contexts to decode the whole trajectory including transitions and rewards in other segments. This reconstruction mismatch may lead to averaged latent contexts so that the policy cannot act correspondingly (see detailed analysis and the ablation study in Appendix A.7.3.) For Markovian contexts baseline (Feng et al., 2022) and POMDP baseline (Hausknecht and Stone, 2015), they can perform relatively well on tasks where the contexts can be inferred from only one step transition (i.e., Cheetah Direction and Cheetah Velocity). However, for more complex tasks where the contexts need more steps to infer, SeCBAD significantly outperforms these two baselines (Feng et al., 2022; Hausknecht and Stone, 2015), which proves that the specially designed joint inference component in SeCBAD is effective and can help improve the performance.

To better understand the proposed algorithm, we conduct a series of ablation studies. In Appendix A.7.2, we study the effects of inaccurate prior on SeCBAD. In Appendix A.7.4, we study the implementation choices on how to use $p(G_t|\tau_{1:t})$. In Appendix A.7.5, we further test SeCBAD against different levels of noises within each segment.

### 4.3 Case Studies on the Segment Structure

In order to provide more insights into the results, we present a case study on Ant Direction in this section. We show the behavior and learned latent along with the inferred segment of a randomly selected episode during testing of the SeCBAD algorithm on Ant Direction in Figure 6.

In the first row, we exhibit the task direction in orange as well as the agent's actual direction in blue. It shows that SeCBAD can rapidly detect and adapt to context changes. We also show the learned latent contexts in the second row and the inferred segment structure in the third row. From the third row, it can be seen that the agent can detect the segment in time as the inferred segment structure closely matches the ground truth segment structure. As for the segment starting from the 430th environment step, the change in goal direction is negligible so that the agent automatically merges

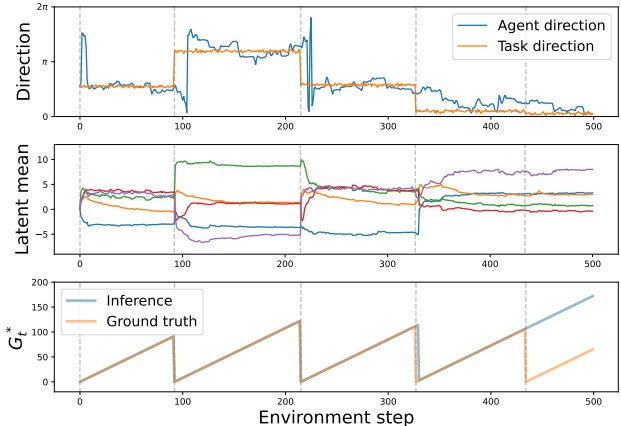

Figure 6: A case study on Ant Direction of SeCBAD. In the first row, we show the behavior of the agent with the goal direction in orange and the actual agent direction in blue. In the second row, we show the learned latent contexts with different colors corresponding to different dimensions. In the last row, we plot the $G_t^*$ inferred by SeCBAD.

the two segments. Since the latent contexts are calculated according to the inferred $G_t^*$, the latent contexts in the second-row change as the segment changes while staying stable within each segment. This proves that the agent can be aware of the changes in unknown contexts and quickly adapt to the changes. We provide case studies of baselines and detailed analysis in Appendix A.7.1.

### 4.4 Bandwith Control Tasks for Real Time Communication

To further illustrate that the proposed LS-MDP setting can boost the deployment of RL in many real-world applications, we test SeCBAD on a real-world bandwidth control task for real-time communications (RTC) (OpenNetLab, 2021) in this section. The most critical goal in RTC is to provide high Quality of Experience (QoE) for users. To achieve this, a bandwidth control module is needed, i.e., the RTC sender needs to decide the bitrate of outstreaming audio/video based on the network status towards the receiver. However, the network conditions constantly change and the changes occur in multiple items, which makes this problem intricate and hard to solve.

In our reinforcement learning formulation, we use a 7-tuple of current network statistics that is visible to the agent as states $s_t$, consisting of sending rate, short-term and long-term receiving rate, loss, and delay. The action $a_t$ is the estimated bandwidth. We provide the detailed settings in Appendix A.8. The latent context $c_t$ here refers to the fluctuated network condition, in this section, we consider $x_t$ as the ground truth bandwidth capacity $C$, and the RTT. In this experiment, we compare our method with VariBAD (Zintgraf et al., 2020), FANS-RL (Feng et al., 2022), vanilla PPO (Schulman et al., 2017a) and PPO-RNN (Hausknecht and Stone, 2015). We also incorporate oracle PPO scores by incorporating the unobservable contexts into the observable states. All the methods are trained for 10 million steps and the shaded area is across 3 random seeds.

As illustrated in Figure 7, SeCBAD achieves better performance than other baselines and is very close to the oracle PPO baseline score. The performances of VariBAD (Zintgraf et al., 2020) and FANS-RL (Feng et al., 2022) are better than PPO-RNN (Hausknecht and Stone, 2015), but SeCBAD outperforms both of these methods. The results suggest that SeCBAD is able to detect and adapt to the varying contexts more rapidly, which allows the policy to precisely control. For detailed descriptions and results, please refer to Appendix A.8.

## 5 Related works

Our work is closely related to **non-stationary RL**, where the transition and reward functions may change over time. Most existing works on non-stationary RL focus on inter-episode non-stationarity (Xie et al., 2021; Chandak et al., 2020a,b; Xie et al., 2022; Sodhani et al., 2021; Alegre et al., 2021; Poiani et al., 2021; Al-Shedivat et al., 2017), some of which adopt contextual MDP as formulation (Hallak et al., 2015). Recently there are some works considering intra-episode non-

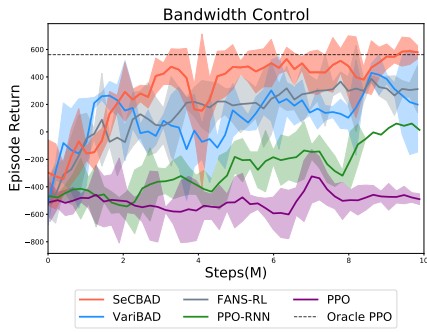

Figure 7: Experiment results on bandwidth control for RTC.

stationarity (Ren et al., 2022; Kamienny et al., 2020; Kumar et al., 2021; Nagabandi et al., 2018; Feng et al., 2022). Ren et al. (2022) assume the latent context to be finite and Markovian, while Feng et al. (2022) assume the latent context is Makovian and the environment can be modeled as a factored MDP. In contrast to existing works on non-stationary MDP, we assume piecewise-stable context with abrupt changes within an episode, which is more realistic and can capture a wide range of real-world applications. To rapidly adapt to dynamic context, the agent needs to continuously perform information gathering behavior. **Bayesian RL** (Duff, 2002; Zintgraf et al., 2020; Fellows et al., 2021) is an elegant framework to optimally tradeoff the exploration and exploitation in an unknown and stationary MDP. As a special type of **Belief MDP** (Kaelbling et al., 1998), Bayes Adaptive MDP (BAMDP) (Duff, 2002) maintains a belief over the environment and uses this belief to augment the state. Our model can be viewed as a special case of belief MDP, where we only maintain belief over latent context to trade off between information gathering and exploitation. To accurately infer belief context, we adopt the **variational inference**, which has been adopted by many existing works in RL for task inference (Rakelly et al., 2019; Zhao et al., 2020; Humplik et al., 2019; Poiani et al., 2021; Zintgraf et al., 2020) or context inference (Xie et al., 2021; Feng et al., 2022; Ren et al., 2022). However, none of these methods suit our setting. We perform joint inference over latent context and segment structure from observed data, so as to remove irrelevant data for more accurate belief context inference. LS-MDP can also be viewed as a special case of **POMDP**. Recently, progress has been made in learning the latent dynamics model (Krishnan et al., 2015; Karl et al., 2016; Doerr et al., 2018; Buesing et al., 2018; Ha and Schmidhuber, 2018; Han et al., 2019; Hafner et al., 2019b,a). Theoretically, it is possible to perform optimally only using recurrent neural networks (RNNs) like Hausknecht and Stone (2015) since the whole history has been taken into consideration. However, it has been shown that (Hafner et al., 2019b) introducing more structured information will significantly enhance the performance. In this paper, we exploit the addtional assumption of LS-MDP to infer the context based on segment structure. The experiments show that SeCBAD can achieve better performance compared with existing methods.

## 6  Conclusion

In this paper, we propose SeCBAD, a **Se**gmented **C**ontext **B**elief **A**ugmented **D**eep RL method to deal with piecewise-stable context in non-stationary environments. Piecewise-stable context is quite common in a wide range of real-world applications. Compared with existing methods, our method can automatically detect the segment structure, which reflects when the context changes abruptly. The detected segment structure can be further used to compute context belief with only relevant observed data. To the best of our knowledge, this is the first method that can model and leverage piecewise-stable context in reinforcement learning to help the agent adapt to environment change. With inferred belief context, our RL agents can quickly detect and adapt to abrupt changes in a gridwold environment and mujoco tasks with piecewise-stable context. For future work, we plan to leverage various deep learning techniques to improve SeCBAD, which includes replacing the GRU encoder by Transformer to better capture the long-term dependency in the input trajectory segment.

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
