# OpenReview forum: "An Adaptive Deep RL Method for Non-Stationary Environments with Piecewise Stable Context"
_NeurIPS.cc/2022/Conference — NeurIPS 2022 Accept_

### Official Review · Reviewer_3SWX · 2022-07-10

**Rating:** 4
**Confidence:** 4
**Soundness:** 3 good
**Presentation:** 3 good
**Contribution:** 2 fair

**Summary:**

This paper provides a new method called SeCBAD (Segmented Contect Belief Augmented Deep RL), to deal with the segment structure of piecewise-stable context. Experiment demonstrates that this method can automatically detect the segment structure, which reflects when the context changes abruptly.

**Questions:**

1. How does this problem compare/relate to distribution shift? Should cite/discuss some distribution shift works.
2. Although it's termed context in this work, the formulation of the method looks quite similar to goal-conditioned RL. Can the authors provide some explanations on how the two are different?
3. I'm a bit confused by the Half-Cheetah Velocity case study. From my understanding, velocity should be the target/reward in this environment. However, it seems to be treated as the context in this paper?
4. Just curious, if this method can detect and adapt to abrupt changes, then can it also adapt to slow and insidious context changes?

**Limitations:**

1. Seems to be a straightforward extension of variational inference framework.
2. Experiment is done in specifically modified environments. Therefore, its objectiveness and fairness remain to be confirmed.
3. Clarity of experiment set up can be improved.
4. The paper claims that 'Piecewise-stable context is quite common in a wide range of real-world applications'. However, the authors did not give a concrete example of this.

**Strengths And Weaknesses:**

Strengths:
The discussed latent situational MDP with piecewise-stable context seems to be potentially useful to model certain stochastic environments. The approach proposed is neat.

Weaknesses:
1. Writing incomplete sentences, e.g. line 5: or assume Markovian context variables for?
2. Lack of theoretical analysis to compare to distribution shift in RL which seems to be a very similar topic.

---

> ### Author Response · Authors · 2022-08-02
> **Response to Reviewer 3SWX**
>
> Thank you for the thoughtful and constructive suggestions! We have taken all the comments into consideration and summarized the responses as follows:
>
> 1. **Discussion on distribution shift**
>
>     To the best of our knowledge, the distribution shift in reinforcement learning mainly refers to the issue in offline RL: the dataset is collected by some behavior policy(s) $\pi_{\beta}$ which is(are) different from the current training policy $\pi$. So training $\pi$ on the dataset induced by other policies $\pi_{\beta}$ may cause some potential problems like overestimation. However, in our case, we do not use a behavior policy to collect the dataset.
>
>     We guess that another possible “distribution shift” maybe refers to the case where the state transition and reward shift around (sorry if we misunderstand). This distribution shift may then refer to non-stationary RL, where the transition and reward functions change over time. (We believe our setting is more related to this “distribution shift”). The discussion of related works about non-stationary RL can be found in Section 5.
>
> ---
>
> 2. **Discussion on goal-conditioned RL.**
>
>     Goal-conditioned reinforcement learning [1] extends MDP with a set of goals $\mathcal{G}$. The reward function is conditioned on the goal as well as state, action: $R: \mathcal{S} \times \mathcal{A} \times \mathcal{G} \rightarrow \mathbb{R}$. We believe there are several major differences between our setting and goal-conditioned RL.
>
>     - For goal-conditioned RL, the MDP transitions are fixed and irrelevant to the goal. For our setting, the transitions are determined by the contexts, and the contexts may change from time to time.
>
>     - For goal-conditioned RL, the goal is known. However, in our case, the contexts are unobservable during the testing (deployment), and we need to infer the contexts.
>
>     - For the goal-conditioned RL setting, the goal is usually fixed within an episode. This is like a special contextual RL setting (Figure 2.a) where the context (goal) lasts for the whole episode and will not affect the transitions.
>
>     [1] Kaelbling, Leslie Pack. "Learning to achieve goals." IJCAI. Vol. 2. 1993.
>
> ---
>
> 3. **Contexts in Half-Cheetah**
>
>     In the half-cheetah velocity environment, the context refers to the "target velocity". For simplicity, let’s denote this "target velocity" (context) $v_g$. The agent’s actual velocity is denoted by $v$. The agent may receive a reward of $- |v - v_g|$ at each timestep. Namely, if the actual velocity is close to the target velocity, then the reward is high. Otherwise, the reward is low. The target velocity evolves in a piecewise stable manner and may change within an episode. So, the agent needs to constantly detect and adapt to the changes to gain high rewards. Please refer to Appendix A.6. "environment details" for more details.
>
> ---
>
> 4. **Generalize to the settings with slow and insidious context changes**
>
>     It’s great to hear such an interesting question! Different real-world applications may have different problem patterns. As shown in Figure 2, the contexts can be constant during the episode (Fig 2.a), be Markovian (Fig 2.b) corresponding to different applications, or be piecewise stable (Fig 2.c). Among these patterns, we find the piecewise stable settings interesting and we, therefore, focus on the setting with piecewise stable contexts, which represents a class of significant real-world problems like congestion control with fluctuated bandwidth and robotics control in changing terrains.
>
>     We believe it’s an interesting setting if the contexts change slowly and insidiously. Though our method is designed for piecewise stable contexts, we guess under certain circumstances it can still work by approximately cutting the changing contexts into different segments (if the probability calculated in Equation 5 indicated so). What’s more, we believe it’s easy to generalize to such a setting by adding some extra components like a learnable context transition module, which is used to model the slow changes within each segment. We may leave this to future works.
>
> ---
>
> 5. **Real-world applications?**
>
>     We are conducting experiments on the bandwidth control problem. However, due to the complexity of this real-world problem and a lack of well-implemented baselines, it takes more time to tune the baselines (such as PPO-RNN, VariBAD) by tuning various hyperparameters, trying different normalization tricks, and feature engineering. Therefore, we may present the experiment results later in the camera-ready version if the paper gets accepted. Nevertheless, we will keep trying during the rebuttal period and update the experiment results for bandwidth control once we get results.

---

> ### Author Response · Authors · 2022-08-08
> **Welcome to see our new experiment on the real-world bandwidth control problem!**
>
> Dear Reviewer 3SWX:
>
> We are glad to let you know that we have conducted a new experiment on a real-world bandwidth control problem!
>
> We add the experiment details and results in Appendix A.8. The results show that SeCBAD achieves better performance than other baselines and performs very close to the oracle PPO baseline (with unobservable contexts as states)! We believe the results further show the significance of our setting and proposed method since we can boost many real-world applications.
>
> We sincerely want to address your further concerns! We believe that we have addressed your previous concerns and all other reviewers' concerns. If you have any additional concerns, please do not hesitate and discuss them with us!
>
> Thank you for your hard work!
>
> Best regards,
>
> The authors

---

### Official Review · Reviewer_ehim · 2022-07-11

**Rating:** 5
**Confidence:** 4
**Soundness:** 2 fair
**Presentation:** 3 good
**Contribution:** 2 fair

**Summary:**

This paper tacles RL on non-stationary MDP. Based on the core assumtion that the environment context is piecewise stable, the authors prospose a Segmented Context Belief Augmented Deep (SeCBAD) RL method, which can jointly infer the belief context and segment structure from observed data. Experiments on a gridworld environment and Mujuco tasks show that the proposed method can quickly detect and adapt to abrupt context changes and outperform existing methods.

**Questions:**

In principle, if the segment of the piecewise stable MDP lasts long enough, variBAD and its variants should be able to fully adapt to each segmented context. For instance, in Figure 4 of the original paper [1], variBAD adapts to a stationary MDP within 1 episode. However, both Figure 5 and Figure 7 show that they learn almost nothing, which is strange.

[1]: Zintgraf, L., Shiarlis, K., Igl, M., Schulze, S., Gal, Y., Hofmann, K., and Whiteson, S. (2020). Varibad: A very good method for bayes-adaptive deep rl via meta-learning. InInternational Conference on Learning Representation (ICLR).


**Strengths And Weaknesses:**

Strengths:

- The paper tackles a relavant problem, non-stationary MDP, and picks a simplified setting based on piecewise stable assumption, which has merits in some real-world scenarios (robotic and congestion control).
- The presentation of the paper is generally clear, despite some typos (e.g. line 5).


Weaknesses:

- **Limited scope**. The contributions of the proposed method depend entirely on the assumption that the non-stationary MDPs are piecewise stable, which inevitably limits the scope of real-world applications, e.g., greenhouse control given stochastic outdoor climate and quantitative trading in unpredictable stock market.

- **Limited novelty**. The proposed algorithm employs the exact same formulation of approximate inference and training objectives of variBAD and makes incremental modifications to the framework to incorporate iterative inference for the segment length. The contributions to both the theoretical and algorithmic aspects of BAMDP are limited.

- **Lack of experiments**. The proposed method is only evaluated on 3 relatively simple environments (1 tabular + 2 MuJoCo) and compared to 1 baseline, variBAD, which is designed for stationary MDP. More baselines targeting non-strationary MDP [1][2] should be included to validate the effectivenss of the proposed method. Also more focused studies such as the robustness of SeCBAD to noise within segmented context could bring more insight to the readers.

Minor issues

- The paper seems a bit rushed. More details of the setup and experiments are needed, for example:
  - How many random seed/trials were performed to produce Figure 5 and 7?
  - What is $G$ defined in line 89?

[1]: Ren, H., Sootla, A., Jafferjee, T., Shen, J., Wang, J., and Bou-Ammar, H. (2022).  Reinforcement learning in presence of discrete markovian context evolution.arXiv preprint arXiv:2202.06557

[2]: Feng, F., Huang, B., Zhang, K., and Magliacane, S. (2022). Factored adaptation for non-stationary reinforcement learning.arXiv preprint arXiv:2203.16582.

---

> ### Author Response · Authors · 2022-08-02
> **Response to Reviewer ehim (Part 2 / 2)**
>
> 4. **Why does VariBAD not perform well in piecewise stable contexts?**
>
>     We believe that the performance of VariBAD is related to the number of segments within an episode rather than the segment length. For VariBAD, the reconstruction term uses the latent to decode the whole episode (Eq9 in VariBAD). In the implementation, they randomly sample a timestep to decode given the latent. If there is only one segment, then the learned latent is correctly trained. However, if there are $n$ segments, then the probability of choosing the correct segment becomes $1/n$, which means that the latent might not be correctly trained.
>
>     When the number of segments $n$ becomes larger, the training signal will become noisy, this may result in an averaged context that mixes up all the contexts in the episode. Then, the learned contexts can no more help to distinguish different segments. To prove this, we add an ablation study on AntDir in Appendix A.7.3. We fix the segment length and adjust the episode length so that the number of segments may vary correspondingly. The results of SeCBAD and VariBAD can be found in Figure 12.For SeCBAD, $n$ has little effect on performance. However, for VariBAD, the performance deteriorates as $n$ grows. The results fit the above analysis and show the significance of joint inference on the segment structure and context belief.
>
> ---
>
> 5. **Other details**
>
>     Thanks for your advice. For previously Figure 5 and 7, we use 3 seeds to produce the results. However, we have added a lot of experiments during the rebuttal period and updated Figure 5 and Figure 7. We still intend to use 3 seeds. But due to time constraints, some experiments are still running. We will update our results as soon as the experiments finish. About $G$ in line 89, $G$ here refers to the segment length that is described in line102 (the next paragraph of line89). We have added a brief introduction of $G$ in line 93 to make it more clear.

---

> ### Author Response · Authors · 2022-08-02
> **Response to Reviewer ehim (Part 1 / 2)**
>
> Thank you for the thoughtful and constructive suggestions! We have taken all the comments into consideration and summarized the responses as follows:
>
> 1. **About the scope: the significance of piecewise stable contexts.**
>
>     Different real-world applications may have different problem patterns. As shown in Figure 2, the contexts can be constant during the episode (Fig 2.a), be Markovian (Fig 2.b) corresponding to different applications, or be piecewise stable (Fig 2.c). Among these patterns, we find the piecewise stable settings interesting and we, therefore, focus on the setting with piecewise stable contexts, which represents a class of significant real-world problems like congestion control with fluctuated bandwidth and robotics control in changing terrains. As agreed by other reviewers (Reviewer PKbi and Reviewer 3SWX), our setting is realistic and useful, and therefore can improve the applicability of RL algorithms to real-world problems. For other context patterns like greenhouse control given stochastic outdoor climate and quantitative trading in the unpredictable stock market, we may leave them to future works.
>
> ---
>
> 2. **About the novelty**
>
>    We believe that the piecewise stable context setting is significant and can boost many real-world applications. This setting proposes unique challenges on context inference that are different from existing works and cannot be solved using existing methods. What’s more, by now, few existing works focus on this setting and it lacks corresponding attention. In our paper, we propose a framework to jointly infer the belief distribution and segment structure and use the context belief augmented state to learn the policy. We believe the proposed SeCBAD is a natural and suitable tool for this setting and hopefully can inspire future works on the same topic.
>
> ---
>
> 3. **About the experiments: We have added more environments and baselines.**
>
>     Thanks for your advice! We have added more environments as well as baselines, and we have rewritten the experiment section. We agree that it will be more informative to compare the proposed methods with non-stationary MDP baselines. According to your advice, we choose FANS-RL [1] to be our baseline among [1] and [3]. We omit [3] because both [1] and [3] are of the same types (both of them focus on non-stationary settings). We believe [1] is more close to our setting since experiments in [3] are focused on low-dimensional environments (4-dimension Cartpole and 12-dimension drone & intersection), and [3] assumes the number of contexts to be finite and treats it as a hyperparameter which needs to be specified before training. Additionally, we add [2] to represent the POMDP method since our LS-MDP can be viewed as a special POMDP. We also add other environments. The results are presented in Figure 5 and Section 4.2 (we rewrite this Section). SeCBAD can still achieve superior performance and sample efficiency compared with all the baselines.
>
>     We provide some insights into the results. For the method assuming that contexts stay the same within an episode, VariBAD uses the learned latent contexts to decode the whole trajectory including transitions and rewards in other segments. This reconstruction mismatch may lead to averaged latent contexts so that the policy cannot act correspondingly (see the next comments for more details).
>
>     For Markovian contexts baseline [1] and POMDP baseline [2], they can perform relatively well on tasks where the contexts can be inferred from only one step transition (i.e., Cheetah Direction and Cheetah Velocity). However, for more complex tasks where the contexts need more steps to infer, SeCBAD significantly outperforms these two baselines [1,2], which proves that the specially designed joint inference component in SeCBAD is effective and can help improve the performance. (Please refer to Section 4.2 for more details).
>
>     We also add other ablation studies to provide more insights into the results. In Appendix A.7.4, we test different approaches to use the inferred $p(G_t|\tau_{1:t})$ when further inferring the contexts.  In Appendix A.7.5, we show that SeCBAD is robust to the context noises within each segment. Hopefully, these results may provide more insights and help to understand our methods.
>
>     [1] Feng, Fan, et al. "Factored Adaptation for Non-Stationary Reinforcement Learning." arXiv preprint arXiv:2203.16582 (2022).
>
>     [2] Hausknecht, Matthew, and Peter Stone. "Deep recurrent q-learning for partially observable mdps." 2015 aaai fall symposium series. 2015.
>
>     [3] Ren, H., Sootla, A., Jafferjee, T., Shen, J., Wang, J., and Bou-Ammar, H. (2022). Reinforcement learning in presence of discrete markovian context evolution.arXiv preprint arXiv:2202.06557

---

> ### Author Response · Authors · 2022-08-08
> **Welcome to see our new experiment on the real-world bandwidth control problem!**
>
> Dear Reviewer ehim:
>
> We are glad to let you know that we have conducted a new experiment on a real-world bandwidth control problem!
>
> We add the experiment details and results in Appendix A.8. The results show that SeCBAD achieves better performance than other baselines and performs very close to the oracle PPO baseline (with unobservable contexts as states)! We believe the results further show the significance of our setting and proposed method since we can boost many real-world applications.
>
> We sincerely want to address your further concerns! We believe that we have addressed your previous concerns and all other reviewers' concerns. If you have any additional concerns, please do not hesitate and discuss them with us!
>
> Thank you for your hard work!
>
> Best regards,
>
> The authors

---

### Official Review · Reviewer_PKbi · 2022-07-16

**Rating:** 5
**Confidence:** 3
**Soundness:** 2 fair
**Presentation:** 3 good
**Contribution:** 2 fair

**Summary:**

Authors investigate the non-stationary RL problems defined as scenario MDP where there exists an unobservable context variable that could change abruptly and unpredictably but remain stable within a stochastic period.  The general idea of this paper is to develop a joint inference framework for the segment structure and the belief context. A variational inference method similar to variational auto-encoder (VAE) is proposed to infer the context given a segment of the past trajectory. An iterative inference method is also proposed for the segment length.  The infered context is then augmented to the observed state variable as inputs to RL algorithms. Experiments on grid world and Mujuco are conducted to verify the performance of the proposed method.

**Questions:**

1. What is the complexity of the proposed algorithms? According to Eq. 5, the posterior of the belief context has to sum over all possible segmentation and if it is the case, then the computational complexity seems to large for larget t.

2. If we look at the motivating example in Fig.1, it seems that the change of the context can be easily detected using the state variables, e.g. we can apply well developed "change point" detection algorithms to the observables to detect the change of the context. Do we really need such a complex method? I would like authors to further comment on this question.

3. In this paper, is the context variable a scaler? If we extend it to a high dimensional latent variable, whether the proposed method is able to solve the problem is not clear.



**Limitations:**

The major concern of this paper is as stated "Authors do not compare the proposed method to an important brunch of research on Partial Observable MDP (POMDP) in the main text and also in the experiment section. "

**Strengths And Weaknesses:**

Strength:

1. RL for non-stationary environment is an important research topic and advanced methods on this topic could potentially improve the applicability of RL algorithms to real world problems.

2. The scenario MDP formulation proposed in this paper is interesting and could be a more realistic formulation than existing Markovian ones. The joint inference method of the segment structure and the belief context is also non-trivial and thus novel, to the best of the reviewer's knowledge.

3. In general the paper is well motivated and clearly written and techincally sound.

Weakness:

1. Authors do not compare the proposed method to an important brunch of research on Partial Observable MDP (POMDP) in the main text and also in the experiment section.

2. The RL part is fairly weak as the proposed method simply augment the learned context to the state and then feed them to any existing RL algorithms.  To me, the whole framework seems a two-stage approach where the connection between the two stages are quite weak.

3. The experiment section is weak. As stated as the motivation of the paper, non-stationarity could be the major issue for RL to be applied to real tasks, and I expect authors to demonstrate the performance of their algorithms on real tasks. However, experiments on real tasks are not presented.

---

> ### Author Response · Authors · 2022-08-02
> **Response to Reviewer PKbi (Part 2 / 2)**
>
> 4. **Why not use existing change point detection methods?**
>
>     We believe that we cannot directly apply change point detection methods in our setting. Let’s take Figure 1 as an example: the bandwidth (the blue curve) represents observable contexts $x$ in our formulation, which are only observable during training. The white circles represent latent contexts $c$ in our formulation, which are unobservable for both training and testing. During deployment, neither of these are observable. Therefore, we cannot directly apply change point detection methods to the contexts.
>
>     In SeCBAD, we use state transitions, actions, rewards, and training-only information $x$ to help infer the latent contexts $c$. Since these variables are all helpful according to the PGM in Figure 2.c which can help the inference process. We do not rely on $x$ since we only use it to calculate the reconstruction loss so that we can still apply our method during deployment.
>
> ---
>
> 5. **Higher dimensional contexts?**
>
>     The context variable is not necessarily a scalar. In the MuJoCo environment Ant-velocity, the contexts are 2d-vectors defining the desired x/y velocity. In our algorithm, we treat both $c$ (the latent contexts) and $x$ (the observable contexts) as high-dimensional variables. So, our algorithm can handle the high-dimensional case.
>
> ---
>
> 6. **Real-world applications?**
>
>     We are conducting experiments on the bandwidth control problem. However, due to the complexity of this real-world problem and a lack of well-implemented baselines, it takes more time to tune the baselines (such as PPO-RNN, and VariBAD) by tuning various hyperparameters, trying different normalization tricks, and feature engineering. Therefore, we may present the experiment results later in the camera-ready version if the paper gets accepted. Nevertheless, we will keep trying during the rebuttal period and update the experiment results for bandwidth control once we get results.

---

> ### Author Response · Authors · 2022-08-02
> **Response to Reviewer PKbi (Part 1 / 2)**
>
> Thank you for the thoughtful and constructive suggestions! We have taken all the comments into consideration and summarized the responses as follows:
>
> 1. **About the experiment part: We have added more environments and baselines.**
>
>     Thanks for your advice! We have added more environments as well as baselines, and we have rewritten the experiment section. We agree that it will be more informative to compare with POMDP methods since the proposed LS-MDP is a special case of POMDP. For the POMDP baseline, we use [2]. We have additionally added another baseline: FANS-RL [1]. [1] represents the class of methods assuming intra-episode context changes and models $c_t$  at each timestep. [2] represents a POMDP method since our LS-MDP can be viewed as a special POMDP. The results are presented in Figure 5 and Section 4.2 (we rewrite this Section). SeCBAD can still achieve superior performance and sample efficiency compared with all the baselines.
>
>     We provide some insights into the results. For the method assuming that contexts stay the same within an episode, VariBAD uses the learned latent contexts to decode the whole trajectory including transitions and rewards in other segments. This reconstruction mismatch may lead to averaged latent contexts so that the policy cannot act correspondingly (see detailed analysis and the ablation study in Appendix A.7.3.).
>
>     For Markovian contexts baseline [1] and POMDP baseline [2], they can perform relatively well on tasks where the contexts can be inferred from only one step transition (i.e., Cheetah Direction and Cheetah Velocity). However, for more complex tasks where the contexts need more steps to infer, SeCBAD significantly outperforms these two baselines [1,2], which proves that the specially designed joint inference component in SeCBAD is effective and can help improve the performance. (Please refer to Section 4.2 for more details).
>
>     We also add other ablation studies to provide more insights into the results. In Appendix A.7.4, we test different approaches to use the inferred $p(G_t|\tau_{1:t})$ when further inferring the contexts.  In Appendix A.7.5, we show that SeCBAD is robust to the context noises within each segment. Hopefully, these results may provide more insights and help to understand our methods.
>
>     [1] Feng, Fan, et al. "Factored Adaptation for Non-Stationary Reinforcement Learning." arXiv preprint arXiv:2203.16582 (2022).
>
>     [2] Hausknecht, Matthew, and Peter Stone. "Deep recurrent q-learning for partially observable mdps." 2015 aaai fall symposium series. 2015.
>
> ---
>
> 2. **About the RL part**
>
>     In our paper, we propose SeCBAD, a framework to learn and use the context belief coherently. For the inference part, we propose to jointly infer the latent contexts and the segment structure, and for the RL part, we augment the state using the inferred belief. We believe this is a natural choice according to the belief MDP and Bayesian Adaptive MDP frameworks. The two components (learn belief / learn belief) form a novel and effective framework to solve a class of real-world applications that is significant yet lacks attention.
>
> ---
>
> 3. **The complexity of the proposed algorithm?**
>
>     In Equation 5, given the inferred $G_t=i$, we only need to calculate the likelihood of observing current $(s,a,r)$ given previous state, action, and inferred contexts, which can be done with $O(1)$ computation. However, one potential drawback of our work lies in Equation 4. To get the full distribution $p(G_t | \tau_{1:t})$ for timestep $t$ may require $O(t)$ computation. When the episode is long, it may require more computation for later timestep. However, we believe that this can be mitigated using many approximation methods. For instance, we can check the segment structure for every $k$ step for long episodes. This can reduce computation to $1/k$ only at the cost of delaying detection for $k$ steps, which is not a big deal for a long episode. We can also design other heuristic methods like truncating the history that is too long ago according to the prior for some problems. We believe that it’s an interesting research topic to study how to reduce the complexity of our setting and we leave this for future work.

---

> ### Author Response · Authors · 2022-08-08
> **Welcome to see our new experiment on a real-world bandwidth control problem!**
>
> Dear Reviewer PKbi:
>
> We are glad to let you know that we have conducted a new experiment on a real-world bandwidth control problem!
>
> We add the experiment details and results in Appendix A.8. The results show that SeCBAD achieves better performance than other baselines and performs very close to the oracle PPO baseline (with unobservable contexts as states)! We believe the results further show the significance of our setting and proposed method since we can boost many real-world applications.
>
> We sincerely want to address your further concerns! We believe that we have addressed your previous concerns and all other reviewers' concerns. If you have any additional concerns, please do not hesitate and discuss them with us!
>
> Thank you for your hard work!
>
> Best regards,
>
> The authors

---

> > ### Comment · Reviewer_PKbi · 2022-08-09
> > **post-rebuttal comment**
> >
> > Thanks for the detailed rebuttal which clarifies part of my concern. I am willing to raise my score.

---

### Official Review · Reviewer_5PEc · 2022-07-17

**Rating:** 6
**Confidence:** 4
**Soundness:** 3 good
**Presentation:** 3 good
**Contribution:** 3 good

**Summary:**

The paper introduces the problem setting where the environment has a “piecewise-stable context” (i.e. the context is constant for some stochastic length of time then changes abruptly to another context several times in an episode), called LS-MDP. Previous works tackle either extremes where either (1) only a single constant context is sampled for the episode or (2) there is a Markovian context at each timestep. The paper then proposed a method, SeCBAD, which performs joint inference for the segment structure (i.e. how long each context occurs for) the belief context given segment structure using variational inference techniques, and combines them in a mixture distribution to marginalize over the possible segment structures. The policy then is learned with the belief context as additional information (instead of conditioning on the exact context).

Experiments were conducted both in a didactic grid-world example, as well as in larger school Mujoco environments, i.e. inferring ideal movement direction for ant locomotion and half-cheetah with changing wind direction. The inferred segment structure is also visualized in the half-cheetah environment to indicate agreement with the underlying context changes.

**Questions:**

Please see my questions below:
1. The experiments would be stronger with additional baselines in addition to the VariBAD baseline. For example, one of the other works assumes intra-episode context changes and models $c_t$ at each timestep. This will help to strengthen the result that the piecewise-stable prior helps over Markovian context type approaches in your environment settings.
2. It would be good to provide empirical data to back up the claim that using $G_t^*$ with the highest probability in $p(G_t|\tau_{1:t}))$ and then using the corresponding belief lead to little performance loss.
3. What is the sensitivity of the training schedule between training the encoder/decoder/policies? In Algorithm 1, it seems that they are all updated at the same pace, but I am wondering if you have also explored for example training the encoder/decoder for more steps relative to the policy optimization step, etc.
4. More hypothetical, not asking for experimental data: Could the context transition also depend on the states reached? E.g. reaching a new terrain, etc. How would your method behave in those types of environments?


**Limitations:**

As mentioned above, the paper did not describe much limitations and potential negative societal impact of their work. Perhaps it is also worth to discuss about the potential sample complexity required to collect enough data to learn the encoder/decoder, or what happens if the underlying MDP has markovian contexts / one context, then would the approach still be able to also solve those situations just as well as other baselines (i.e. if there is a mismatch in the MDP assumption and the method).

**Strengths And Weaknesses:**

**Originality**: The paper was effective in contextualizing its problem formulation (pun intended) in the sea of non-stationary RL works, while clarifying that LS-MDP is a special case of POMDP. Given these additional assumptions in the problem setting, then we can exploit them in the solution. The proposed approach applies well-known tools like variational inference methods for this problem setting, but there is merit in showing that these existing tools can provide significant gains over methods that have more general/different problem assumptions (such as VariBAD).

**Quality**: The paper mainly provides empirical results to analyze the proposed method to demonstrate that it can incorporate the piecewise-stable context for adapting to abrupt environmental changes. However, the limitations of the method was not discussed in detail, nor was there a section on potential negative societal impacts the work.

**Clarity**: Overall the paper is relatively clear to read, with helpful conceptual diagrams for the method. There are several minor comments for improving the draft:

1. In the abstract, second sentence seems incomplete.
2. Figure 4: should label the y-axis as ‘time step’ or something along that lines to help quickly understand how to read the plots. Based on the last paragraph on page 7 I was also expecting a plot for the vanilla inference behaviour but I do not see that in there.
3. For the G_t plots, perhaps it’s also informative to plot the oracle $G_t^*$ value (i.e. the perfect zig-zag lines) to help compare that with the estimated $G_t^*$ by the proposed algorithm

**Significance**: The paper proposes a relevant problem setting and demonstrated that their method can yield noticeable improvement. There are still a lot of architecture improvements to the method (as mentioned about using Transformer instead of GRU in the recurrent encoder), and more likely also new approaches for the given problem setting.

---

> ### Author Response · Authors · 2022-08-02
> **Response to Reviewer 5PEc (Part 2 / 2)**
>
> 6. **Could the context transition also depend on states?**
>
>     It’s great to hear such an interesting question! We believe this is a very appealing setting. In our paper, we mainly focus on the setting where the contexts are completely exogenously given so that the agent needs to identify and adapt to the completely exogenously contexts. Taking the robotics control in changing terrain as an instance, if the states refer to the sensor readings (velocity, angular velocity, etc) like in MuJoCo, then we can assume the contexts are independent of states. If the states refer to the visual inputs (e.g., camera), then the contexts might be dependent on the states (we can see the road). In this case, it might be possible for us to learn a $\hat{c} = f(s)$ to help derive the context information from the state.
>
>
>     However, it might be possible that, if the context transition depends on the states, the contexts are not exogenous but become something controllable and observable. Then the problem may reduce to an ordinary MDP rather than a non-stationary MDP, or LS-MDP in our setting. We believe the key challenges may change in this new setting, but our algorithm may still work because we can still infer the contexts from historical observations since the state sequence still contains necessary information. In this case, we can generalize our method by adding some extra components to help infer the contexts from states, which can be explored in future work.

---

> > ### Comment · Reviewer_5PEc · 2022-08-07
> > **Thank you for your response**
> >
> > I have read the responses from the authors. Thank you for clarifying the limitations of the work, as well as strengthening the experiments by providing additional baselines that assume intra-episode context changes (FANS-RL) and a more general POMDP approach (PPO-RNN) in Section 4.2. The Appendix A.7.4 section also clarified the usage of $p(G_t|\tau_{1:t})$ as well as requested. I am increasing my score rating to a weak accept.

---

> > > ### Author Response · Authors · 2022-08-08
> > > **Thank you for raising the score!**
> > >
> > > Thank you for raising the score! We sincerely appreciate your support for our work, and we will keep polishing our work in the future.

---

> > > ### Author Response · Authors · 2022-08-08
> > > **Welcome to see our new experiment on the real-world bandwidth control problem!**
> > >
> > > Dear Reviewer 5PEc:
> > >
> > > We are glad to let you know that we have conducted a new experiment on a real-world bandwidth control problem!
> > >
> > > We add the experiment details and results in Appendix A.8. The results show that SeCBAD achieves better performance than other baselines and performs very close to the oracle PPO baseline (with unobservable contexts as states)! We believe the results further show the significance of our setting and proposed method since we can boost many real-world applications.
> > >
> > > If you have any additional concerns, please do not hesitate and discuss them with us!
> > >
> > > Thank you again for your hard work!
> > >
> > > Best regards,
> > >
> > > The authors

---

> ### Author Response · Authors · 2022-08-02
> **Response to Reviewer 5PEc (Part 1 / 2)**
>
> Thank you for the thoughtful and constructive suggestions! We have taken all the comments into consideration and summarized the responses as follows:
>
> 1. **Potential limitations of our work**
>
>     Thanks for your advice. One potential limitation of SeCBAD is that accurate segment inference, i.e., to get the full distribution $p(G_t | \tau_{1:t})$ for timestep $t$, may require $O(t)$ computation. When the episode is long, it may require more computation for later timesteps. However, we believe that this can be mitigated using many approximation methods. For instance, we can check the segment structure for every $k$ step for long episodes. This can reduce computation to $1/k$ only at the cost of delaying detection for $k$ steps, which is not a big deal for a long episode. We can also design other heuristic methods like truncating the history that is too long ago according to the prior for some problems. We believe that it’s an interesting research topic to study how to reduce the complexity for our setting and we leave this for future work.
>
> ---
>
> 2. **Writing issues:**
>
>     Thanks! We have modified the second sentence in the abstract and completed Figure 4 and the corresponding description in Section 4.1. What’s more, we have adjusted Figure 6 to help make it more informative according to your advice.
>
> ---
>
> 3. **Additional baselines:**
>
>     According to your advice, we have added more environments and additional baselines to our works: FANS-RL [1] and PPO-RNN [2]. [1] represents the class of methods assuming intra-episode context changes and models $c_t$  at each timestep. [2] represents a POMDP method since our LS-MDP can be viewed as a special POMDP. The results are presented in Figure 5 and Section 4.2 (we rewrite this Section). SeCBAD can still achieve superior performance and sample efficiency compared with all the baselines.
>
>     We provide some insights into the results. For the method assuming that contexts stay the same within an episode, VariBAD uses the learned latent contexts to decode the whole trajectory including transitions and rewards in other segments. This reconstruction mismatch may lead to averaged latent contexts so that the policy cannot act correspondingly (see detailed analysis and the ablation study in Appendix A.7.3.).
>
>     For Markovian contexts baseline [1] and POMDP baseline [2], they can perform relatively well on tasks where the contexts can be inferred from only one step transition (i.e., Cheetah Direction and Cheetah Velocity). However, for more complex tasks where the contexts need more steps to infer, SeCBAD significantly outperforms these two baselines [1,2], which proves that the specially designed joint inference component in SeCBAD is effective and can help improve the performance. (Please refer to Section 4.2 for more details).
>
>     We also add other ablation studies to provide more insights into the results. In Appendix A.7.5, we show that SeCBAD is robust to the context noises within each segment. Hopefully, these results may provide more insights and help to understand our methods.
>
>     [1] Feng, Fan, et al. "Factored Adaptation for Non-Stationary Reinforcement Learning." arXiv preprint arXiv:2203.16582 (2022).
>
>     [2] Hausknecht, Matthew, and Peter Stone. "Deep recurrent q-learning for partially observable mdps." 2015 aaai fall symposium series. 2015.
>
> ---
>
> 4. **How to use the inferred $p(G_t|\tau_{1:t})$?**
>
>     Thanks for your advice! We add an ablation study in Appendix A.7.4. Since there are $t$ possible choices for $p(G_t|\tau_{1:t})$ in timestep $t$, it's hard to directly use the full distribution. One option is to use the $G_t$ with the highest probability. Another option is to sample $G_t \sim p$ to approximate the full distribution. We show an ablation study in Figure 13 and the performances of the two options are very close. So we claim that using $G_t^*$ with the highest probability may lead to little performance loss in Section 3. (Please refer to Appendix A.7.4 for more details).
>
> ---
>
> 5. **The training schedule between the encoder/decoder/policies?**
>
>     We use different training schedules between the VAE part (encoder & decoder) and the policy part. We use two Adam optimizers with different learning rates. The VAE optimizer uses a learning rate of 1e-3, and the policy optimizer uses a learning rate of 7e-4. We tuned the parameters and we observed that using different training schedules has better results.  We have added these details to our paper.

---

### Author Response · Authors · 2022-08-02
**Overall Response**

We sincerely appreciate all reviewers’ and ACs’ time and efforts in reviewing our paper. We thank you all for the insightful and constructive suggestions, which helped further polish our paper. We would like to clarify our Motivation and contributions as follows:

**Motivations:**

Adapting to variations of unknown environment contexts is one of the key challenges in deploying RL to many real-world applications. In this paper, we focus on a significant case where contexts evolve in a piecewise stable manner. The key challenge lies in how to infer the unobservable contexts with segment information unknown.

**Our contributions can be summarized as follows:**

- To the best of our knowledge, we are the first to propose SeCBAD, a Segmented Context Belief Augmented Deep RL method, to solve problems with piecewise stable contexts, which are of great application significance yet lack attention and remain unsolved.

- We propose a novel method that can jointly infer the segment structure as well as the belief of latent contexts. The inferred belief can then be leveraged to augment the state following the belief MDP framework.

- We empirically test the effectiveness of the proposed methods.

    - The motivating example on the grid world shows that inference with segment structure as SeCBAD can provide more accurate results than vanilla inference(which is used in many baselines).

    - Through the control experiments, we show that SeCBAD achieves superior performance and sample efficiency.

---

Here is a summary of our updates:

**Additional Experiments:**

- **[New update!]**: We add a real-world bandwidth control experiment! We compare SeCBAD with other baselines to show the significance of the proposed setting and algorithm in real-world applications. (Appendix A.8)

- We add several MuJoCo environments and we have 5 different tasks. Now we can then evaluate the proposed algorithm more comprehensively! (Section 4.2)

- As suggested by Reviewer 5PEc and Reviewer ehim, we add a baseline considering intra-episode non-stationarity which assumes the contexts can change within an episode. We choose FANS-RL [1] as our baselines.

- As suggested by Reviewer PKbi, we add a POMDP baseline since our setting can be regarded as a special case of POMDP.

- As suggested by Reviewer ehim, we add an ablation study on the number of segments within an episode. We provide some analysis of the results which can provide some insights into why existing methods assuming constant contexts may fail in our setting. (Appendix A.7.3)

- As suggested by Reviewer 5PEc, we add an ablation study on how to use the learned $p(G_t|\tau_{1:t})$ to further back up the implementation choice in our paper. (Appendix A.7.4)

- As suggested by Reviewer ehim, we add an ablation study on the robustness of noise within segmented context to show that SeCBAD is robust and can handle the in-segment noises. (Appendix A.7.5)

- Some experiments are still running. We will upload the results once they finish.

---

**Writing:**

As for writing, we owe many thanks to all reviewers’ extremely helpful suggestions. During the rebuttal, we revise our manuscript

- We rewrite the experiment section to provide more experiments and analysis.

- Some ablation studies are moved to Appendix (Appendix A.7.3 - Appendix A.7.5) due to the page limitation.

- We also fix other issues and highlight all the modifications in blue.

---

We sincerely thank all reviewers for discussing with us! We have uploaded the revised paper and commented on all the reviews. We hope we have addressed all of your concerns and will continue to polish our paper in our revision. Thanks again for your comments, and we sincerely wish you could reconsider your score.

---

### Meta-Review · Area_Chair_QWEi · 2022-08-26

**Recommendation:** Accept
**Confidence:** Less certain

**Metareview:**

This paper proposes an algorithm to deal with non-stationarity in RL, in settings where a latent context variable changes abruptly at discrete points in time (in contrast to previous work that focuses either on settings where the context is constant over an episode, or may change at each timestep)

This is very much a borderline paper, with reviews in the "weak reject" to "weak accept" range. Summarizing the main concerns raised by reviewers:
1. Some writing issues
2. Lack of comparison to other baselines
3. Lack of experiments on closer to real-world tasks
4. Limited novelty
5. Limited applicability (piecewise stable context)

The authors made significant improvements during the discussion period, in particular by:
1. Fixing the main writing issues
2. Adding comparisons to more baselines, in particular a POMDP algorithm and an algorithm for non-stationary RL
3. Adding a bandwidth control task that is closer to a real world setting (+ also additional MuJoCo tasks for a more comprehensive evaluation)

In my opinion, concerns #1-2-3 have thus been addressed in a satisfactory manner.

Regarding #4 (novelty), it is indeed somewhat limited since this work is very close to the prior VariBAD algorithm, and can be seen as an incremental improvement to make it handle intra-episode non-stationarity. That being said, I consider that tackling this "piecewise stable context" setting is a novel and valuable contribution, since it is definitely an under-explored setting in the current literature, and can have some interesting real world applications.

Concern #5 remains since this work is indeed specific to the piecewise stable context setting, and thus may (1) not be useful when there is no intra-episode non-stationarity, and (2) not work well in cases where the context changes frequently / continuously. However, I would not consider it a fatal flaw since I do believe this piecewise stable setting to be interesting and relevant.

I also note that the only reviewer recommending rejection (3SWX) did not reply to the authors' response (and neither did they update their review) to explain why they were still leaning towards rejection in spite of the above-mentioned improvements.

In conclusion, I believe the updated submission is now meeting the bar for publication, as it shows strong empirical results in a novel setting that is relevant to (some) real world applications.

**Award:**

No

---

### Decision · Program_Chairs · 2022-09-14

Accept